# Physiologic Cyclical Load on Inguinal Hernia Scaffold ProFlor Turns Biological Response into Tissue Regeneration

**DOI:** 10.3390/biology12030434

**Published:** 2023-03-11

**Authors:** Giuseppe Amato, Roberto Puleio, Giorgio Romano, Pietro Giorgio Calò, Giuseppe Di Buono, Luca Cicero, Giovanni Cassata, Thorsten Goetze, Salvatore Buscemi, Antonino Agrusa, Vito Rodolico

**Affiliations:** 1Department of Surgical, Oncological and Oral Sciences University of Palermo, 90127 Palermo, Italy; 2Department of Pathologic Anatomy and Histology, IZSS, 90129 Palermo, Italy; 3Department of Surgical Sciences, University of Cagliari, 09123 Cagliari, Italy; 4CEMERIT-IZSS, 90129 Palermo, Italy; 5Institut für Klinisch-Onkologische Forschung Krankenhaus Nordwest, 60488 Frankfurt/Main, Germany; 6Department PROMISE, Section Pathological Anatomy University of Palermo, 90127 Palermo, Italy

**Keywords:** arteries, inguinal protrusion disease, muscle, muscle growth factors, neo-angiogenesis, neo-myogenesis, neo-nervegenesis, nerve, nerve growth factors, regenerative scaffolds, tissue degeneration, tissue regeneration, vascular growth factors, veins

## Abstract

**Simple Summary:**

A 3D scaffold developed for inguinal hernia repair has been designed to overcome the multiple incongruences of currently used flat meshes. Conventional hernia implants, positioned in a high-motile environment, are static, need fixation and, instead of healing the degenerative source of hernia disease, produce a granuloma of low-quality scar tissue. On the contrary, the new 3D scaffold, endowed with centrifugal expansion, avoids fixation, reacts dynamically to the motile impulses of the groin and attracts tissue growth factors, promoting the development of newly formed tissue structures. Despite being made of the same polypropylene material as conventional flat meshes, the dynamic behavior of the 3D scaffold that moves in tune with inguinal movements proves that modifying the design and kinetic attitude turns the biological response into tissue regeneration.

**Abstract:**

Surgical repair of groin protrusions is one of the most frequently performed procedures. Currently, open or laparoscopic repair of inguinal hernias with flat meshes deployed over the hernial defect is considered the gold standard. However, fixation of the implant, poor quality biologic response to meshes and defective management of the defect represent sources of continuous debates. To overcome these issues, a different treatment concept has recently been proposed. It is based on a 3D scaffold named ProFlor, a flower shaped multilamellar device compressible on all planes. This 3D device is introduced into the hernial opening and, thanks to its inherent centrifugal expansion, permanently obliterates the defect in fixation-free fashion. While being made of the same polypropylene material as conventional hernia implants, the 3D design of ProFlor confers a proprietary dynamic responsivity, which unlike the foreign body reaction of flat/static meshes, promotes a true regenerative response. A long series of scientific evidence confirms that, moving in compliance with the physiologic cyclical load of the groin, ProFlor attracts tissue growth factors inducing the development of newly formed muscular, vascular and nervous structures, thus re-establishing the inguinal barrier formerly wasted by hernia disease. The development up to complete maturation of these highly specialized tissue elements was followed thanks to biopsies excised from ProFlor from the short-term up to years post implantation. Immunohistochemistry made it possible to document the concurrence of specific growth factors in the regenerative phenomena. The results achieved with ProFlor likely demonstrate that modifying the two-dimensional design of hernia meshes into a 3D outline and arranging the device to respond to kinetic stresses turns a conventional regressive foreign body response into advanced probiotic tissue regeneration.

## 1. Introduction

The conventional concept of modern prosthetic inguinal hernia repair implies the deployment of flat meshes to cover the herniated groin [1]. Flat meshes are static, do not interact with the high-motile environment of the groin and, being fastened with sutures or tacks, hinder the kinetics of the inguinal area, thus resulting in being painfully unphysiological [2,3]. Immune reaction to hernia meshes activates a foreign body response, forming a granulomatous fibrotic plaque that incorporates the implant [4]. This phenomenon represents the so-called “reinforcement” of the weakened groin but is inconsistent with the purpose of healing the degenerative roots of the disease [5,6,7,8,9,10,11,12,13]. Indeed, with reference to the degenerative genesis of inguinal hernia disease, the goal of treatment should be to stop degeneration and activate tissue regeneration [14]. These physiological and pathogenetic incongruences seem to underlie frequent complications occurring after conventional inguinal hernia repair [15,16,17,18]. Despite the above-mentioned discrepancies, no concrete solution has been advocated to overcome these problems. Based on recent evidence on functional anatomy and pathogenesis, a new concept of cure has been developed to manage inguinal protrusions. It concerns a newly designed 3D device, a scaffold named ProFlor, whose dynamic and regenerative features seem to resolve all specific issues deriving from the use of flat meshes in inguinal hernia repair [19,20,21]. The proprietary centrifugal expansion of ProFlor allows for a permanent obliteration of the hernia defect without need for fixation. The 3D device is compressible on all planes and, on being introduced into the hernial opening, moves in compliance with the cyclical load of the inguinal structures. The dynamic compliance of ProFlor has been demonstrated to turn the biological response into a regenerative effect, activating specific tissue growth factors to induce the development of newly formed connective tissue, muscles, vessels and nerves: a true regenerative feature [22,23,24,25,26,27,28,29,30]. This paper aims to summarize the results of the histological and immunohistochemical evidence gathered through the analysis of tissue specimens excised from patients who underwent inguinal hernia repair with ProFlor. The data highlighted herewith demonstrate that the use of the same material, but modifying the structure of current static hernia devices from a flat design to a 3D outline provided by inherent dynamic responsivity, is crucial to achieve a completely different biological response. Specifically, there is the development of newly formed highly specialized tissue elements and no granulomatous, low-quality scar tissue.

## 2. Material and Methods

The Ethics Commission of the Medical Board of the Land Hessen—Germany ethically approved the research (Approval Number: FF32/2013). The study was performed in accordance with the Declaration of Helsinki for experiments involving humans. All participants taking part in the study signed informed written consent.

The study involved patients who had already undergone primary groin hernia with the 3D hernia scaffold ProFlor that for different reasons required re-intervention in the same inguinal region. In particular, the biopsies were removed in patients that needed surgery in the groin for various pathologies, e.g., hydrocele in the groin already operated for inguinal hernia, hernia protrusion overlooked in the primary repair or recurrence. In these patients, biopsy samples were removed from the 3D scaffold for histological examination. The investigation utilized the same biopsy samples used for previous studies removed from 15 individuals between 2010–2015 [20,21,22,23,24]. The 3D scaffold ProFlor, produced by Insightra Medical Inc. Clarksville, TN—USA, is designed with a 3D core with a cylindrical multilamellar outline, with reinforced margins, and is manufactured with low weight with large porous polypropylene, and has a thickness of 0.15 mm. The lamellas are 15 mm thick with edges that have been rolled and welded with ultrasound and measuring 0.8 mm in thickness. Two polypropylene rings connect the lamellar structure of the 3D core to achieve a flower-like outline. With this configuration, the 3D core is compressible on all surfaces and comes in 2 sizes: 25 mm (composed of six petals) and 40 mm (with eight petals) (Figure 1A). A flat mesh is connected at the center of the 3D structure on one surface of the core. The connected flat mesh has different sizes depending on the dimension of the 3D core: a 60 mm circular one on the 25 mm scaffold, while the 40 mm scaffold has a rounded 70 mm mesh. Another version of the 3D scaffold, the ProFlor E (extended), utilized for laparoscopic approach or to repair large hernia protrusions, features an oval flat mesh, 80 × 100 mm. When positioned in the defect, this flat mesh counterfaces the peritoneum. The proprietary centrifugal expansion of ProFlor permits a fixation-free obliteration of the hernia opening. The intrinsic dynamic behavior of ProFlor allows for contraction and relaxation in tune with inguinal movements.

The tissue samples gathered for the investigation were excised from 3 patients between 3 and 10 weeks post implantation (short-term—ST); from 5 patients between 3 and 5 months post-operation (mid-term—MT); from 4 patients between 6 and 12 months post implantation (long-term—LT); and in the last 3 patients more than one year after the operation (extra-long-term—ELT). All biopsy samples were removed from the anterior aspect of ProFlor, which faces the fascia of the external oblique muscle and has no connection to other inguinal structures (Figure 1B,C). Therefore, no host native tissue could be included in the removed tissue specimen.

### 2.1. Histological Study

All excised tissue samples, fixed in neutral-buffered 10% formalin, were immersed in paraffin wax. Then, 4-μm-thick slices were prepared and kept until usage at room temperature. Histological analysis, focused on the plain detection of muscle, vascular and nervous structures, was performed with hematoxylin and eosin (H&E). Azan Mallory stain (Bio-Optica Milano S.p.A. Milano, Italy) was utilized to highlight the connective structure and the presence of muscle tissue in the 3D scaffold.

### 2.2. Immunohistochemistry

Following dewaxing, for antigen retrieval, a solution of Tris EDTA (pH 9.0) heated the slides for 20 min at 96 °C. A solution of 3% *w*/*v* hydrogen peroxide in methanol for 30 min was utilized to block endogenous peroxidase activity. Slides were processed for 15 min with Background Sniper. Then, all sections were incubated with primary antibodies (NSE, CD31, SMA, VEGF, NGF, NGFR p75) at room temperature (RT) for 1 h. Details on the immunohistochemical antibodies utilized for the research are shown in Table 1.

After incubation, the slides were washed out three times for five minutes with phosphate-buffered saline (PBS), then treated at room temperature for 30 min with biotinylated immunoglobulin (LSAB, Dako Agilent, Santa Clara, CA USA). After twice cleansing with PBS for 5 min, the slides were incubated at room temperature for 60 min with streptavidin, horseradish peroxidase conjugate. After washing out 3 times with Tris-buffer saline (TBS), the slides were incubated with chromogen 3-3′-diaminobenzidine tetrahydrochloride (DAB; Dako Agilent, Santa Clara, CA, USA). Sections were then cleansed in running water and counterstained with Mayer’s hematoxylin, following dilution for 1 min in 0.035% TBS. In the case of positivity, the DAB-reaction established a brown precipitate. Specific primary antibodies were replaced in tissue sections with normal goat serum or PBS and were used as negative controls. Tissue specimens with specific tissue markers (vascular, nervous, or muscle tissue) were used as a positive control. All immune-stained slides were examined at a 100× magnification by means of Leica DMR microscope equipped with a Nikon DS-Fi1 digital camera and image capture software (NIS Basic Research Nikon software).

Monoclonal primary antibody NSE (LSBio) was applied at 1:100 dilution for 60 min to highlight evidence of newly developed nervous structures in the 3D device.

## 3. Results

Biopsy samples and related histologic specimens were blindly examined by two pathologists in respect to the timing of excision. The series of figures included herewith, shown according to the postoperative stages of the biopsies, are ordered, first showing the specific tissue growth factors found followed by microphotographs, which highlight the stage of development of the related tissue elements. Details of the histological and immunohistochemical evidence are summarized in Table 2.

### 3.1. Evidence of Angiogenetic Growth Factors and Vascular Structures in ProFlor

Concerning the evidence of vascular tissue components, slides of the specimens removed after 3–5 weeks post-implantation (short-term—ST) showed clearly detectable vascular endothelial growth factor (VEFG)-modulated neoangiogenesis (Figure 2A) and a noteworthy CD31-mediated activity finalized to the structural development of the vessels (Figure 2B). Concerning the ingrowth of the muscle layer of the vessels, a limited presence of SMA-positive elements could be observed (Figure 2C). As a result of this multifaceted imprinting of growth factors, analysis of the H&E slides showed numerous clusters of newly ingrown, still immature, vascular elements that likely supported the new tissue structures in development. Negligible inflammatory infiltrate composed of lymphocytes and histiocytes in a surround of fibroblast proliferation and development of connective elements also characterized this stage (Figure 2D). In the samples excised 3–4 months post-implantation (mid-term—MT), VEFG presence decreased (Figure 3A) while CD31 evidenced clearly increased vessel density (Figure 3B). In this time frame, increased presence of smooth muscle actin (SMA)-induced smooth muscle development in veins and arteries was also detectable (Figure 3C) and the tissue specimens stained with H&E showed an increased number of angiogenetic clusters in advanced stage of development in complete absence of inflammatory reaction against the polypropylene structure of the 3D scaffold (Figure 3D). Six to eight months post-operation (long-term—LT), a reduced presence of VEGF-positive elements could be seen (Figure 4A); however, CD31-mediated endothelial ingrowth was manifested (Figure 4B). Contextually, SMA immunohistochemistry highlighted the development of the muscle layer composing the arterial media, proving that structural maturation was already finalized (Figure 4C). Overall, the H&E-stained slides demonstrated the presence of mature, well-structured arteries and veins that were seen in the newly ingrown tissue of ProFlor (Figure 4D). In the subsequent phase beyond 1 year post-operation (extra-long-term—ELT), a limited number of VEFG-stained elements could be observed in the newly ingrown tissue of ProFlor (Figure 5A), while CD31- and SMA-positive cells were clearly detectable (Figure 5B,C). In more detail, in the context of no inflammation, H&E slides showed abundant vascular structures complete in all layers: the intima, muscular and adventitia (Figure 5D).

### 3.2. Evidence of Myogenetic Growth Factors and Muscular Structures in ProFlor

Concerning the presence of myogenetic growth factors in the biopsies removed in the short-term after placement of the 3D scaffold, a striking number of nerve growth factor (NGF)-positive cells in the polypropylene fabric of the 3D scaffold were reported (Figure 6A). As a consequence, in this early phase, multiple spots of myocytes in an embryonic stage of development could be identified within the unstructured but well-hydrated connective tissue composing the substrate of the newly ingrown tissue (Figure 6B). In detail, in the newly developed tissue of ProFlor prominent nucleoli, vesiculated nuclei and moderate basophilia were observed; these elements constitute the distinctive trait of the initial stage of myocytic development. In the samples gathered in the mid-term (MT), an increasing number of NGF positive cells were present in the polypropylene fabric of the 3D scaffold (Figure 6C). In this phase, myocytic development was evidently intensified in quality and quantity and a higher quantity of myofibril spots and broad areas of muscle bundles dispersed in well-organized connective tissue were highlighted adjacent to the fabric of the 3D hernia device. In detail, muscle bundles showed small nuclei, spindle fashioned outline and eosinophilic cytoplasm, and presented in the phase of progressive development, representing the typical feature of the myocyte evolutive phase (Figure 6D). In LT, several NGF-positive cells were evident within the texture of ProFlor, indicating an intense progression of the myogenetic impulse (Figure 7A). The effect of such a regenerative feature exerted by NGF was documented by the presence of large areas of mature muscle bundles within the 3D scaffold distributed among well-perfused connective tissue (Figure 7B). In the following ELT postoperative period, several NGF-positive cells were highlighted in the polypropylene structure of ProFlor (Figure 7C). As a consequence, the muscle elements evidenced in ProFlor increased in quantity and, being arranged in bundles, appeared as standard mature muscle elements. These myocytic bundles were disseminated in a surround of viable and woven connective tissue whose outline appeared arranged following defined lines of force. Highly magnified images taken in this long-term stage showed striated muscle components with the typical spindle-shaped outline of myocytes with hyperchromic small nuclei and eosinophilic cytoplasm fully resembling normal human muscle bundles (Figure 7D).

### 3.3. Evidence of Nervous Structures in ProFlor

Concerning the presence in the short-term (ST) of the neurogenic growth factor NGFRp75, this nerve development mediator, unlike NGF, was seen in limited quantities throughout the structure of ProFlor (Figure 8A). At this point in time, multiple spots of immature nervous structures and related myelin sheathes in the stage of initial development immersed in well-perfused connective tissue could be seen within the 3D scaffold (Figure 8B). In the mid-term (MT), in the examined biopsy samples, an increased quantity of NGF-Rp75 positive elements was clearly seen (Figure 8C). This was associated with the presence, dispersed in well-ordered connective tissue, of many large nervous structures in the stage of progressive development, enfolded by well-represented myelin sheathes adjacent to the structure of ProFlor (Figure 8D). In the long-term (LT), compared to previous postoperative stages, an increased amount of NGFRp75-positive spindle cells were regularly identified in the cytoplasm and on the membrane of the cellular elements adjacent to the structure of the 3D hernia device (Figure 9A). In this stage, many mature nervous structures were found adjacent to the fabric of the ProFlor device. In detail, the nervous structures documented within ProFlor in the LT had clearly increased in number, demonstrating an effective and solid development of both myelin and axons, thus resembling a typical human nerve in all constituting elements. (Figure 9B) In the extra-long-term (ELT), microphotographs showed a significant number of NGFRp75-positive elements close to the polypropylene fibers of ProFlor (Figure 9C). As a consequence, tissue specimens removed from the 3D scaffold in the ELT confirmed the consolidated development of the nervous structures ingrown within ProFlor. Indeed, highly magnified images demonstrated a typical outline of mature, well-structured nerves, constituted by bundles of nervous fibers assembled in fascicles and enfolded by viable myelin sheathes, capable of isolating the core of the neural elements from the external environment (Figure 9D).

## 4. Discussion

Prosthetic reinforcement of the herniated groin with flat meshes mostly made of polypropylene is considered the gold standard for the cure of inguinal hernia disease [31,32,33]. Conventional hernia meshes are dynamically passive and do not participate in the cyclical load of the groin, one of the most mobile areas of the body. To avoid migration, meshes are fastened to the myotendineal structure of the groin with sutures or tacks. However, the need for fixation represents a physiological incongruence, as fastened meshes hinder groin movements and increase postoperative pain as well as the rate of adverse events [34,35,36]. Another controversial aspect of conventional herniorrhaphy with flat meshes concerns consistency with the pathogenesis of the disease. Rationally, to defeat the degenerative roots of inguinal protrusion disease, stopping tissue degeneration and inducing regeneration of the wasted structures should be the scope of the cure. Nevertheless, foreign body granuloma induced by flat and static meshes do not appear to be a regenerative but rather a regressive feature. Concerns about such pathogenetic inconsistency is further heightened by considering that the typical fibrotic incorporation of the mesh is uncontrolled and may also include the highly sensitive nerves running in the inguinal canal. This event, occurring in up to 20% of patients, is responsible for the worst postoperative complications after herniorrhaphy: life-wasting chronic pain syndrome [15,16,17,18,36].

Aiming to improve such controversial treatment results, a new concept has been conceived for a more physiological and pathogenetic coherent method. It concerns a 3D scaffold, ProFlor, designed with a multilamellar shape, available for open and laparoscopic approaches and made of polypropylene, similar to conventional hernia implants [37,38,39]. Compressible on all planes with recoil memory, the 3D core of the scaffold takes advantage of its natural centrifugal expansion to achieve a permanent fixation-free obliteration of the hernia defect. Regenerative scaffolds are widely used in scientific research and can be defined as 3D structures created to enable development and proliferation of new cellular elements when implanted in patients [40]. The low weight and large porous polypropylene structure of the 3D scaffold ProFlor provide high permeability, which, as confirmed in literature, is essential for the passage of growth factors and migration of cells to achieve tissue regeneration [41].

The regenerative scaffold ProFlor has been developed to dynamically interact with the motile environment of the groin, compressing and relaxing in compliance with the inguinal musculature, being therefore fully in line with the physiology of this area of the body. While being produced with polypropylene material, its biological response is in no way comparable to the foreign body granuloma typical of flat hernia meshes made of the same material [25]. Indeed, referring to a long series of investigations, the development up to complete structural finalization of newly formed, highly specialized tissue elements, comparable to those that typically compose the inguinal barrier, has been demonstrated within ProFlor [26,27,28,29]. This probiotic effect constitutes an unprecedented feature for an inguinal hernia device. In detail, the presence of muscle elements in ProFlor has been seen gathered in bundles in a setting of viable connective tissue, innervated by fully functional, newly ingrown mature nervous structures and perfused by well-structured arteries and veins. As demonstrated above in the results, the development of newly formed vascular structures within the 3D scaffold is mediated by specific growth factors, having a definite timely efficacy. In the short-term (ST), VEGF triggers the early stage of vascular development. In this stage, newly formed vessels start to develop as documented by the infrequent and CD31- and SMA- positive vascular elements. After a few months, even if in the ingrowing stage, a diffuse vascular network is already functional within the 3D scaffold. In this phase, VEFG content progressively reduces while CD31 positivity is highlighted, shaping the structure of the vessels. Contextually, widespread positivity for SMA demonstrates the mature muscle structure of arteries and veins: in this stage, the abundance of an SMA-active area can be detected in the 3D device. This demonstrates an evident escalation in quantity of actin myofibrils enclosed in the ingrown vascular elements, contributing to the development of a thick muscular layer composing the vessels. After 6/8 months, VEFG is slightly detectable, CD31 remains active while SMA still contributes to stabilizing the muscular components of veins and arteries. Beyond this period, the progressive development of mature vascular elements able to fully satisfy the perfusion of sophisticated tissue components ingrown in the 3D scaffold, in particular muscles and nerves, is already finalized.

On the other hand, development of skeletal muscle is subject to stimulation, development and differentiation of myocytic ancestors. Cytokines and specific growth factors constituting the microcellular environment of myogenic precursors activate the development of myofibrils [42]. In the initial phase of development, myocytes start to produce neurotrophic elements and related specific receptors. This implies the presence of NGF, tyrosinekinase receptors (TrkA and TrkB) and the p75-neurotrophin receptor (p75NTR) [43,44]. NGF is a neurotrophin that plays an important role in the life of nervous elements [45]. Several studies also demonstrate that NGF is fundamental in managing critical metabolic phenomena in other tissue structures. Two different types of receptors control NGF signaling: the tropomyosin-related kinase (Trk) receptor and the p75 neurotrophin receptor (NGFRp75) [46]. It has also been demonstrated that neurotrophin/NGFR p75 in vitro induces survival of myotubes as well as myogenic differentiation, while its role in the repair of damaged myocytes has been demonstrated in vivo [43,44].

Nerve growth factor receptor (NGFR) is also referred to as p75(NTR) due to its molecular mass and capability to bind not only NGF, but also to other neurotrophins, including brain-derived neurotrophic factor (BDNF), neurotrophin-3 (NTF3), and neurotrophin-4 (NTF4) at low affinity [47]. During regeneration and development of peripheral nerves, Schwann cells show higher levels of the neurotrophin receptor p75(NTR). Neurotrophins are key mediators inducing myelination of the peripheral nervous system. Cosgaya et al. demonstrated that the absence of functional p75(NTR) inhibits myelin formation [47]. Furthermore, p75(NTR) receptor induces the enhancement of myelin formation by endogenous BDNF, whereas TrkC receptors, well-known receptors for neurotrophins that regulate neuronal survival, are responsible for the inhibition of neurotrophin-3. Therefore, p75(NTR) and TrkC receptors have opposite effects on myelination. Cosgaya established a model for the actions of endogenous neurotrophins and their receptors through myelination [47]. During glial proliferation, elongation, and ensheathment, the levels of NT3 reduce, while TrkC and p75(NTR) remain stable. Therefore, the positivity to NGF and NGFRp75 highlighted in the biopsy samples excised from ProFlor at different periods likely explains the presence of newly formed muscle and nervous structures in progressive phases of cellular development, significantly making up the newly ingrown tissue within the 3D scaffold.

In light of the above, in all probability, the proprietary regenerative behavior that characterizes ProFlor is in compliance with the steady movements induced by the physiologic cyclical load of the groin muscles. As previously mentioned, while being made of the same material, ProFlor differs in design from traditional hernia meshes. This difference gives rise to a distinctive reaction to dynamic stresses: conventional static meshes are flat and, being passive, do not interact with the recipient; on the contrary, the 3D scaffold responds in tune to dynamic impulses of the inguinal barrier where it is firmly positioned. Briefly, conventional hernia flat meshes do not move and produce low-quality foreign body granulomas while ProFlor moves in synchrony with the groin and allows for the regeneration of the tissue components wasted by the degenerative injury of hernia disease. A limitation of this research, which warrants attention in future studies, pertains to the limited patient cohort. Although the findings are statistically significant, the number of biopsies (15) could have been increased with additional samples. However, it took several years to acquire enough biopsies to proceed with the study, as such opportunities were rare.

## 5. Conclusions

For inguinal hernia repair, in contrast to the mainstream, old-fashioned and controversial treatment concept finalized to “reinforce” the groin with the stiff fibrotic incorporation of flat static meshes, the dynamic responses of ProFlor to the physiologic cyclical load during inguinal movements likely demonstrate that, using the same material, it is sufficient to modify the design and physical aspect of the mesh to achieve a totally different biological response. Dynamic compliance, regenerative behavior, fixation-free placement and effective obliteration of the hernia opening characterize this novel device. First in the field of inguinal hernia therapy, ProFlor would appear to embody an innovative technology that combines resistance to pathogenetic insult with a new type of a regenerative effect able to restore the integrity of a fully functional inguinal barrier. The unique technology of ProFlor likely represents a turning point for an advanced cure of inguinal hernias that aims to reduce adverse events and improve patients’ quality of life.

## Figures and Tables

**Figure 1 biology-12-00434-f001:**
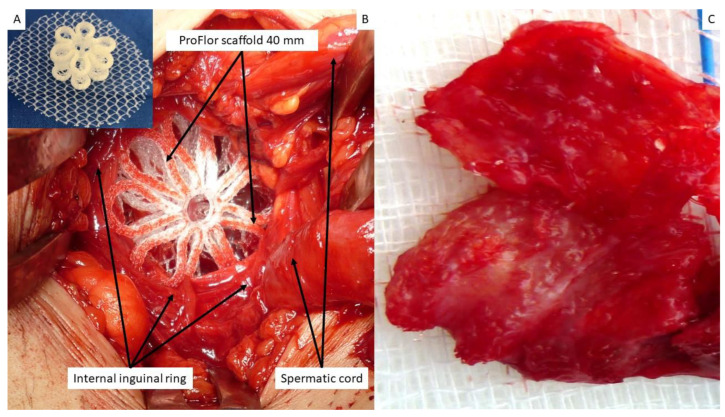
(**A**) The multilamellar 3D outline of the 25-mm-sized ProFlor with the rounded flat mesh connected at the center of the posterior surface. (**B**) At operation, a 40-mm-sized 3D scaffold ProFlor compressed by the musculature of the internal inguinal ring fully obliterates an indirect hernia defect. (**C**) Biopsy specimen excised 8 months post-implantation from the anterior aspect of the 3D scaffold. The removed specimen appears to be made up of viable muscle tissue; the polypropylene fabric of the scaffold is no longer identifiable. Of note, the excised tissue sample corresponds to the surface of the scaffold seen in (**A**), which solely faces the external oblique fascia. The fascia does not contain muscles; therefore, no host native tissue contaminates the removed specimen.

**Figure 2 biology-12-00434-f002:**
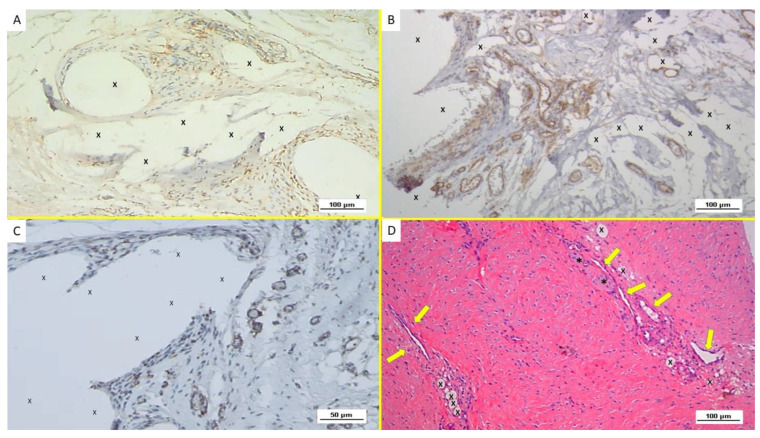
Biopsy specimen excised from ProFlor 6 weeks post-implantation (short-term—ST). (**A**): Several VEGF-positive angiogenetic clusters composed of vascular elements in the stage of development (brown spots) close to the polypropylene fabric of the 3D scaffold (X). VEGF 100X—(**B**): Immunohistochemistry in biopsy sample excised from ProFlor 4 weeks post-implantation (short-term—ST). Many clusters of vasculo-endothelial elements in the early stage of development (stained in brown) throughout the polypropylene fabric of ProFlor. CD31 100X—(**C**): biopsy sample excised from the 3D scaffold 5 weeks post-implantation (short-term—ST). Rare actin-positive vascular structures (stained in brown) close to the fabric of the 3D hernia device (X). SMA 200X—(**D**): Biopsy specimen excised from ProFlor 5 weeks post-implantation (short-term—ST). Several angiogenetic clusters composed of vascular elements in the stage of development close to the polypropylene fabric of the 3D scaffold (X) with negligible inflammatory response. An elongated nervous structure in development is also detectable (*). EE 100X.

**Figure 3 biology-12-00434-f003:**
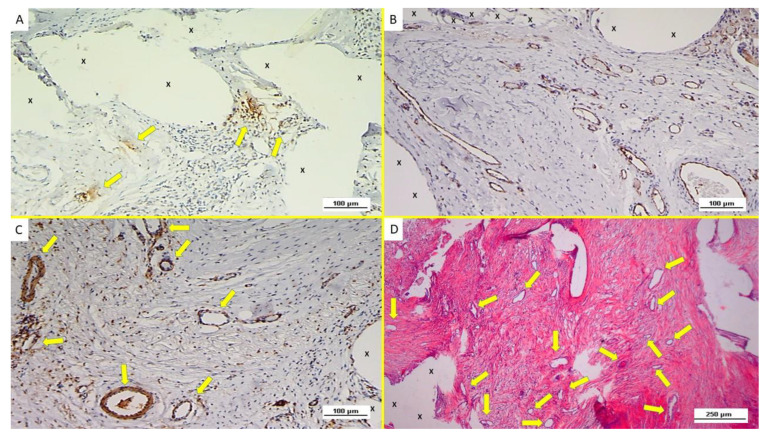
Immunohistochemistry of biopsy specimen excised from ProFlor 4 months post-implantation (mid-term—MT). (**A**): VEGF-stained vascular elements (yellow arrows) in the stage of development close to the polypropylene fabric of the 3D hernia device (X). VEGF 100X. (**B**): Biopsy specimen excised from ProFlor 3 months post-implantation (mid-term—MT): many vasculo-endothelial structures (stained in brown) in the advanced stage of development throughout the polypropylene fibers of the 3D scaffold (X). CD31 100X. (**C**): Biopsy specimen excised from ProFlor 4 months post-implantation (mid-term—MT): increased evidence of SMA-induced smooth musculature development in arteries and veins was detectable. SMA 100X (**D**): Biopsy sample removed from ProFlor 5 months post-implantation (mid-term—MT): several vascular structures in the advanced stage of development (yellow arrows) close to the polypropylene fabric of the 3D scaffold (X). No inflammatory reaction detectable. EE 50X.

**Figure 4 biology-12-00434-f004:**
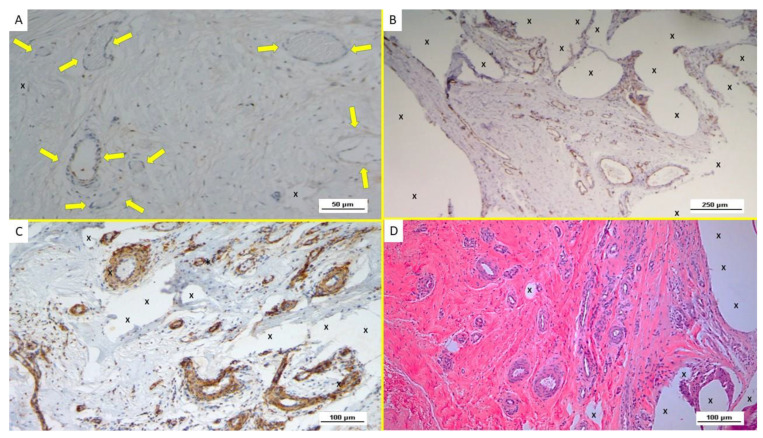
Immunohistochemistry of biopsy specimen excised from ProFlor 7 months post-implantation (long-term—LT). (**A**): VEGF-positive elements compose the endothelial layer of vascular structures (yellow arrows) close to the polypropylene fabric of the 3D hernia device (X). VEGF 200X. (**B**): Biopsy specimen excised from ProFlor 7 months post-implantation (long-term—LT): abundant mature endothelial structures (stained in brown) of several vessels dispersed throughout the polypropylene scaffold of ProFlor. CD31 50X. (**C**): Biopsy specimen removed from ProFlor 8 months post-implantation (long-term—LT): noteworthy amount of thick muscle layers composing arteries (brown targeted spots) and veins (brown elongated elements) highlighted within the polypropylene fabric of the 3D scaffold (X). SMA 100X. (**D**): Biopsy sample removed from ProFlor 9 months post-implantation (long-term—LT): abundant well-structured arteries and veins (red/white targeted spots) newly developed within the polypropylene fabric of the 3D scaffold (X) in absence of inflammatory elements. EE 100X.

**Figure 5 biology-12-00434-f005:**
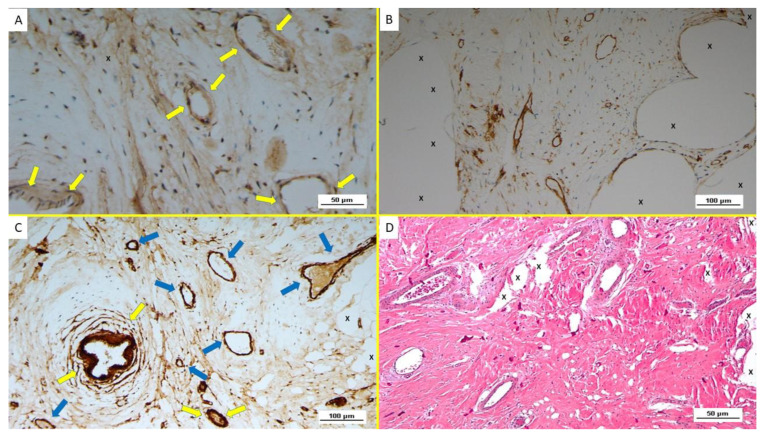
Biopsy specimen excised from ProFlor 28 months post-implantation (extra-long-term—ELT). (**A**): limited amount of well-structured VEGF positive venous elements in the stage of advanced development (yellow arrows). VEGF 200X. (**B**): Biopsy specimen excised from ProFlor 24 months post-implantation (extra-long-term—ELT): several mature vascular elements showing well-structured endothelium (stained in brown) highlighted throughout the polypropylene structure of ProFlor (X). CD31 100X. (**C**): Biopsy specimen excised from ProFlor two years post-implantation (extra-long-term—ELT): several SMA-positive muscle layers of arteries (yellow arrows) and veins (blue arrows) close to the polypropylene fabric of ProFlor (X). SMA 100X—(**D**): Biopsy sample removed from ProFlor three years post-implantation (extra-long-term—ELT): several mature vascular structures close to the polypropylene fabric of the 3D scaffold (X). No inflammatory reaction detectable. EE 200X.

**Figure 6 biology-12-00434-f006:**
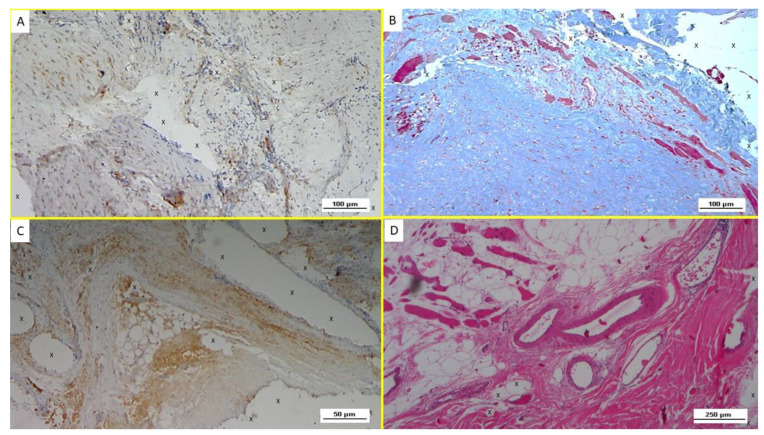
(**A**): Biopsy specimen excised from ProFlor 4 weeks post-operation (short-term—ST): noteworthy amount of NGF positive cells (colored in brown) close to the polypropylene fabric of the 3D scaffold (X). NGF 100X. (**B**): Biopsy of ProFlor removed 5 weeks post implantation (ST): viable connective tissue (colored in blue) enclosing multiple spots of newly ingrown muscle elements and muscle bundles (stained in red) in the early stage of development close to the polypropylene fabric of the 3D scaffold (X). AM 100X—(**C**): Tissue specimen removed from ProFlor 4 months post-operation (mid-term—MT): increasing amount of NGF-positive cells stained in brown surrounded by the polypropylene fabric of the 3D scaffold (X). NGF 200X—(**D**): Biopsy specimen excised from the ProFlor 5 months post implantation (mid-term—MT). Several bundles of muscle elements in an advanced development stage are highlighted in red in the lower right section of the microphotograph close to the implant fibers (X) and surrounding large arterial and venous elements (white-red targeted spots). In the left superior section of the image, further muscle elements stained in red are dispersed, surrounded by adipocytes (white circular spots). EE 50X.

**Figure 7 biology-12-00434-f007:**
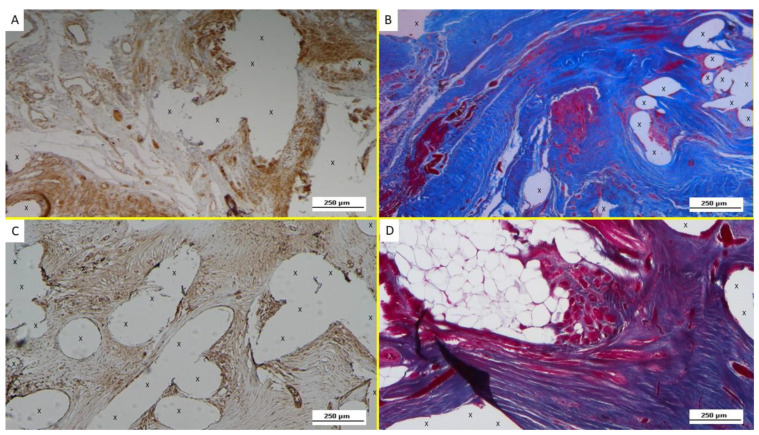
(**A**): Biopsy specimen excised from ProFlor 8 months post-operation (long-term—LT): several NGF-positive cells close to the polypropylene fibers (X). The white/brown targeted spots on the left upper section of the image indicate the presence of many vascular elements. NGF 50X—(**B**): Biopsy sample excised from ProFlor 7 months post implantation (long-term—LT)—(**A**): large areas of mature muscle bundles (stained in red) within the polypropylene fibers (X) of the 3D scaffold distributed in a context of well-perfused connective tissue (stained in blue). On the left upper margin, several vascular structures are clearly detectable. AM 50X—(**C**): Biopsy specimen excised from ProFlor 36 months post-operation (extra-long-term—ELT): evidence of several NGF-positive cells colored in brown and surrounding the polypropylene structure of ProFlor (X). NGF 50X—(**D**): Biopsy specimen excised from the ProFlor 24 months post implantation (extra-long-term—ELT). Abundant mature muscle bundles stained in red close to a cluster of adipose tissue (white spots) surrounded by a large number of connective fibers close to the polypropylene fabric of the 3D scaffold (X). AM 50X.

**Figure 8 biology-12-00434-f008:**
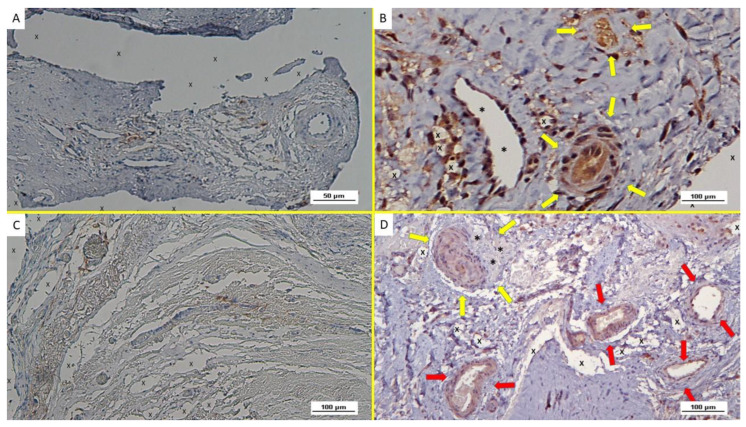
(**A**): Biopsy specimen excised from ProFlor 4 weeks post-operation (short-term—ST): evidence of NGFRp75-positive cells colored in brown close to the polypropylene fabric of the 3D scaffold (X). On the right margin of the specimen, a middle-sized arterial structure is also evident. NGFRp75 200X—(**B**): Biopsy from ProFlor removed 5 weeks post implantation (short-term—ST): close to the implant fibers (X), close to a large vein (*) two nervous structures are present, showing axons and related myelin sheath in stages of development (yellow arrows). NSE 100X—(**C**): Biopsy specimen excised from ProFlor 4 months post-operation (mid-term—MT): evidence of NGFRp75-positive elements (stained in brown) in the context of ProFlor (X). NGFRp75 100X—(**D**): Microphotograph of biopsy excised from the 3D scaffold 5 months after implantation MT: presence of a large nervous structure in the stage of intermediate maturation (yellow arrows) and well represented myelin sheath (*) among vascular structures (red arrows), close to the implant fibers (X). NSE 100X.

**Figure 9 biology-12-00434-f009:**
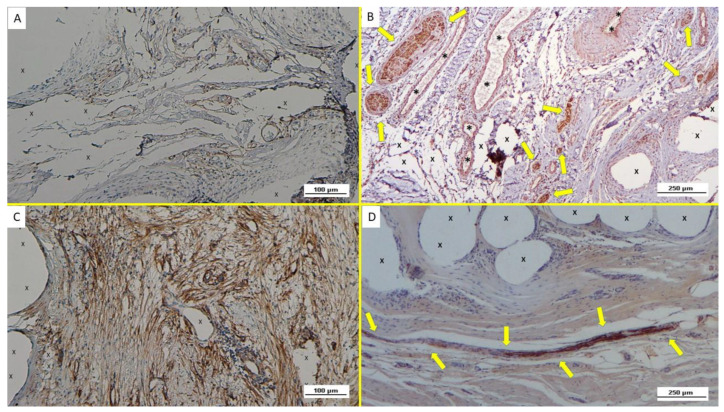
(**A**): Biopsy sample excised from ProFlor 8 months post-operation (long-term—LT): compared to the previous postoperative stage, an increased number of NGFRp75-positive elements close to the polypropylene structure of the 3D scaffold (X) is evident. NGFRp75 100X—(**B**): Biopsy taken from the 3D implant seven months post implantation (long-term—LT): several well-formed nervous structures (yellow arrows), all mature components surrounded by connective tissue, vascular structures (*) and close to the polypropylene fabric of the 3D scaffold (X) NSE 50X—(**C**): Biopsy specimen resected from ProFlor 24 months post-operation (extra-long-term—ELT): the microphotograph shows a significant presence of NGFRp75-positive elements colored in brown close to the polypropylene fibers of the 3D scaffold (X). NGFRp75 100X—(**D**): Microphotograph of biopsy taken from the 3D implant 36 months post implantation (extra-long-term—ELT): elongated mature nervous element (yellow arrows) enfolded in well-structured myelin sheath close to the polypropylene structure of the 3D scaffold. (X). 50X.

**Table 1 biology-12-00434-t001:** Immunohistochemistry antibodies used for processing biopsy specimens.

Antibody	Clone-Code	Source	Dilution
NSE	LS-B14144 (LLSBio)	Rabbit polyclonal	1:100
CD31	JC70A (DAKO)	Mouse monoclonal	1:50
SMA	1A4 (DAKO)	Mouse monoclonal	1:100
VEGF	26503 (R&D)	Mouse monoclonal	1:50
NGF	Ab52918 (Abcam)	Rabbit monoclonal	1:100
NGFp75	Sc 13577 (Santa Cruz)	Mouse monoclonal	1:100

**Table 2 biology-12-00434-t002:** Stages of cellular development and growth factors evidenced in ProFlor over time.

	Growth Factors					Tissue Structures		
	VEGF	CD31	SMA	NGF	NGFRp75	Vessels	Muscles	Nerves
**Short-term**	+++	++	+	++	+	Evolving	Evolving	Evolving
**Mid-term**	++	+++	++	+++	++	Advanced	Advanced	Advanced
**Long-term**	+	+++	+++	+++	+++	Mature	Mature	Mature
**Extra-long-term**	+	+++	+++	+++	+++	Consolidated	Consolidated	Consolidated

Growth factors evidence: + = limited ++ = noteworthy +++ = abundant.

## Data Availability

All data supporting the reported results are available upon request from the corresponding author.

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
