# Peer review of "Physiologic Cyclical Load on Inguinal Hernia Scaffold ProFlor Turns Biological Response into Tissue Regeneration"

_biology, 2023, doi:10.3390/biology12030434_

Round 1
Reviewer 1 Report
This is a very important and timely manuscript.The 3D scaffold Proflor is a very interesting concept but it does not sound well accepted.Why? There is a need to better describe the device with comprehensive illustrations. The indications for implantation at large and more specifically for this series of biopsies shall be given.A summary table would be a great support to highlight the importance of this manuscript.The size of the biopsies is limited and the authors probably made the best use of them.Several stainings in histology were conducted in addition to immunohistochemistry,this is remarkable.A red picrosirius staining might have been important to precise the development of collagen and type of collagen fibres together with a possible sinusoidal morphology.I am aware that the size of the biopsies is more than small but investigations in transmission electron microscopy are essential to confirm the developments of nervous elements.The iconography is acceptable but confusing and a table sumarizing the results is essential.In the conclusion,the authors shal better tell what this series of explant analyses will change in their surgical practice.The names of the authors of reference 28 to 31 shal be provide.What means dynamically passive?
Reviewer 2 Report
1. Line 14-23, simple summary? I think the important thing is only abstract, please delete the simple summary.
2. The abstract should be broadened to give additional quantitative results.
3. As your abstract's final sentence, include a "take-home" message.
4. Reorder keywords based on alphabetical order.
5. What is the novel of the present study? It works have been widely studied in the past. Nothing something really new in the present form. The lack of novel seems to make the present study like to replication/modified study. The authors need to detail their novelty in the introduction section. It is a major concern for rejecting this paper.
6. To underline the study gaps that the newest research tries to fill, it is crucial to explain the merits, novelty, and limits of earlier studies in the introduction.
7. In line 61, the authors specifically mention ProFlor scaffold? Any specific reasons? It is the best available in the market today? Or what?
8. Before mentioned ProFlor scaffold, the authors need to explain the scaffold in general before specific mentioned the brand. It is such a jumping explanation as a present form. For this purpose, the authors encouraged to incorporated the literature as follows: Level of Activity Changes Increases the Fatigue Life of the Porous Magnesium Scaffold, as Observed in Dynamic Immersion Tests, over Time. Sustainability 2023, 15, 823. https://doi.org/10.3390/su15010823
9. To help the reader grasp the study's workflow more easily, the authors could include more visuals to the materials and methods section in the form of figures rather than sticking with the text that now predominates.
10. What is the baseline of patient selection? Is there any protocol, standard, or basis that has been followed? It is unclear since the patient is very heterogeneous with a small number. The resonance involved impacts the present result makes this study flaws. One major reason for rejecting this paper.
11. Additional information about tools used, such as the maker, country, and specification, should be included.
12. Error and tolerance of experimental tools used in this work are important information that needs to be explained in the manuscript. It is would use as a valuable discussion due to different results in the further study by other researcher.
13. The inaccuracy and tolerance of the experimental equipment used in this inquiry are critical details that must be included in the article.
14. Outcomes must be compared to similar past research.
15. Overall, discussion in the present article is extremely poor. The Authors must extend their discussion and make a comprehensive explanation. Just not simply mention the results with brief explanation.
Round 2
Reviewer 1 Report
I must be more specific in my comments. Let me try to provide some explanations:
1) The 3D scaffold ProFlor is not well accepted as I did find one publication only outside G Amato's circle:CK Palival et al .Minimal dissection..IOSR J Dental Med Sci 2016.
2) The description of the device is inaccurate .I visited Ref 28 and 31:the photos are of better quality but there is no detail of the construction.How the fibers are assembled? Diameters? Visceral side?
Peritoneal side?
3) I confirm that it is not easy to process biopsies because of the size.It is however not mentionned where they are coming from.Visceral or peritoneal side.I am still suspicious about the development of nervous elements.
4) The iconography is interesting but the quality is borderline.The colllagen fibers are difficult to observe and they do not show any sinusoidal structure likely to maintain the softness of the tissues.Please add scales to better demonstrate your findings.
5) There is no reference about the concept ''dynamically passive''.Has this heernia prosthesis been tested in animals? How can the authors propose a concept based upon adverse events.
This manuscript needs a major revision.
Should you need a hernia prosthesis,let me tell you that it is premature to select this device.
Reviewer 2 Report
Followed comments as reviewer response in the revised version is given.
1. Porosity is one of the important aspect in scaffold design since it impacing on the rate of degradation and degeneration. The authors encouraged to highlight it. Also, relevan refesence needs to incorporated as follows: The Effect of Tortuosity on Permeability of Porous Scaffold. Biomedicines 2023, 11, 427. https://doi.org/10.3390/biomedicines11020427
2. After line 447, the limitation of the present study needs to be added at the end of the discussion section before entering the conclusion section.
3. In the conclusion, please explain the further research.
4. Literature from the last five years should be enriched to reference.
5. The authors occasionally created paragraphs in the entire document that were just one or two phrases long, which made the explanation difficult to understand. To make their explanation into a longer, more thorough paragraph, the authors should expand it. It is advised to use at least three sentences in a paragraph, with one serving as the primary sentence and the others as supporting phrases.
6. The manuscript needs to be proofread by the authors since it has grammatical and language issues.
7. Graphical abstract is encouraged to provide in submission after review.
Author Response
Please see the attchment

Round 3
Reviewer 1 Report
Sent previously. You can publish it now.
Reviewer 2 Report
Good job to the authors in their effort.